# Utilizing the predictable binding kinetics of DNA-PAINT to denoise super-resolution images

George Sirinakis [1] ✉, Edward S. Allgeyer [1], Jennifer H. Richens [1], Jeanne Lefévère-Laoide [2], Ewa K. Paluch [2] & Daniel St Johnston [1] ✉

DNA-Point Accumulation for Imaging in Nanoscale Topography (DNA-PAINT) is a versatile super-resolution technique that relies on the predictable binding kinetics between fluorescent imager strands and docking strands attached to target proteins. This makes DNA-PAINT particularly suitable for multiplexing and quantitative applications, but its performance is often limited by spurious signals from non-specific binding of imager strands. Here we describe a method to remove these non-specific binding events using a statistical test to distinguish between DNA-specific and non-specific interactions. To demonstrate the method, we imaged mosaic epithelial tissues in *Drosophila melanogaster* egg chambers and showed that >90% of non-specific and otherwise indistinguishable signal in the super-resolved images can be removed. This denoising improves the quality of DNA-PAINT super-resolved images and is essential for accurate measurements of spatial relationships and protein quantification.

DNA-Point Accumulation for Imaging in Nanoscale Topography (DNA-PAINT) has emerged as a powerful and straightforward Single Molecule Localization Microscopy (SMLM) technique[1–3]. DNA-PAINT exploits the transient but repeatable binding between freely diffusing oligonucleotides attached to fluorophores (imager strands) and complementary oligonucleotides attached to proteins of interest (docking strands) to produce blinks of fluorescent light, which can then be localized to reveal the position of the target protein with sub-diffraction resolution[1].

Since DNA-PAINT relies on DNA-binding kinetics to produce blinking events and not on the photo-physics of the fluorophores, a broader selection of fluorophores may be used, which simplifies multi-colour imaging[1,4]. In addition, since the fluorophores are not directly attached to the target proteins, but freely diffusing in the imaging solution, DNA-PAINT circumvents the problem of photobleaching, and under optimized imaging conditions that minimize docking strand photodamage can achieve higher labelling densities and resolution down to 5 nm[5–7].

However, the presence of freely diffusing fluorophores during imaging can also produce non-specific localization events that appear as noise in the super-resolved images, which reduces image quality, obscures signal of interest and poses a substantial problem for any subsequent analysis steps[8]. An extreme example of this is the nucleus where the abundance of genomic DNA can lead to excessive numbers of non-specific hybridization events. Several methods have been developed to deal with this type of non-specific binding, which either rely on blocking e.g. with salmon sperm DNA[9] or reducing the hybridization affinity by introducing left-handed DNA oligomers[10]. Besides the nucleus, non-specific binding is also present in other parts of the cell, where it is hypothesized to originate from electrostatic interactions between the imager strands and positively charged cellular molecules. Previous work focused on ways to reduce the frequency of non-specific binding events by engineering docking strands with multiple complementary motifs to the imager strands, enabling imaging with lower imager strand concentrations[5,11]. These optimized conditions can offer a sizeable, 5- to 10-fold, reduction in the non-specific binding frequency, but do not eliminate non-specific binding events and additional post-processing steps are required to exclude them, especially for applications that require multiplexing and/or higher resolution[12]. Previous approaches focused on empirical

---

[1]The Gurdon Institute & the Department of Genetics, University of Cambridge, Cambridge, UK. [2]Department of Physiology, Development and Neuroscience, University of Cambridge, Cambridge, UK. ✉e-mail: gs519@cam.ac.uk; ds139@cam.ac.uk

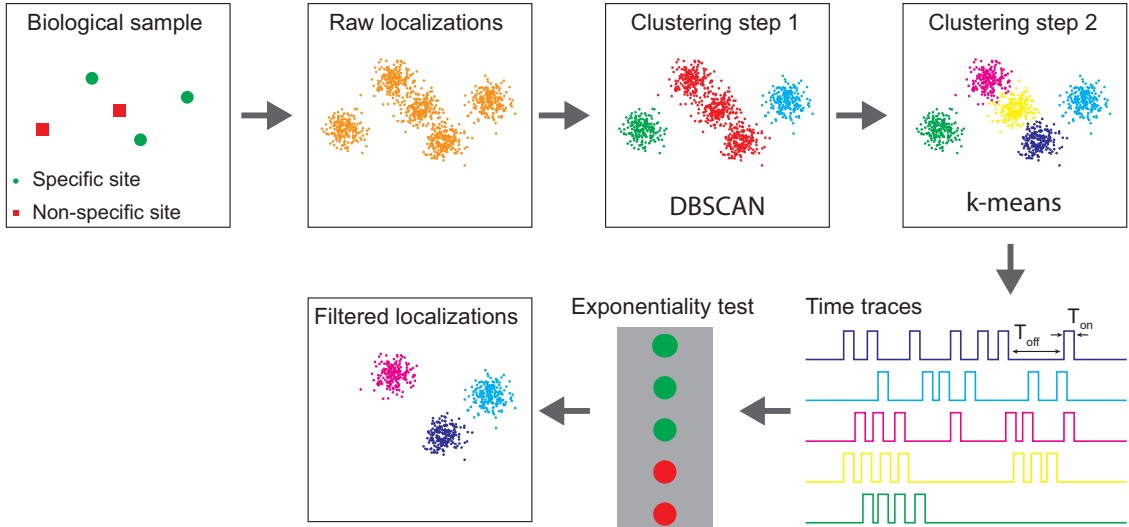

**Fig. 1 | Overview of the denoising method.** Localizations are first clustered based on density by the DBSCAN algorithm followed by a k-means clustering step that partitions these clusters into localization-precision-sized clusters. For each cluster, a time trace representing the binding activity is constructed and the time between binding events is measured. A hypothesis test for exponentiality is performed on the distribution of the times between binding events and clusters with traces that pass the test are filtered as specific.

methods to detect non-specific interactions with very distinct characteristics e.g. non-repetitive binding or bursts in binding activity, but did not take full advantage of the predictable binding kinetics between the imager and docking strands[12,13].

Here we develop a denoising algorithm that exploits the fact that the times separating real binding events between imager and docking strands closely fit an exponential distribution, whereas spurious time traces do not, and propose a filtering method based on a statistical test for exponentiality to distinguish between non-specific and specific DNA-binding events. Our method employs a two-step clustering process that makes it applicable not only to isolated clusters, but also to more crowded environments where target proteins may coalesce into larger structures.

We applied our method to *Drosophila melanogaster* egg chamber epithelial tissues, where we imaged the cell adhesion protein, E-cadherin, and showed that >90% of background signal from non-specific binding can be filtered out while retaining >98% of the specific signal. Our denoising approach improves the quality of DNA-PAINT images, but more importantly, can be combined with recent advances in measuring protein amounts and multiplexing to enable sub-cellular proteomics with single-molecule sensitivity and nanoscale resolution.

## Results

### Clustering of SMLM data

The steps needed to apply the denoising method are illustrated in Fig. 1. DNA-PAINT relies on the transient hybridization between a docking oligonucleotide and a freely diffusing imager oligonucleotide to produce blinking events. These events are subsequently localized to reveal the position of the docking strand with a precision primarily determined by the number of photons collected during binding. Due to the stochastic nature of this process, the times between hybridization events are expected to follow an exponential distribution. This characteristic exponential behaviour is independent of the number and/or spatial distribution of the docking strands, and binding events from a single or several docking strands can be grouped into clusters of any size or shape, as long as the blinks remain well separated in time[2]. Given that the docking strands are attached to proteins with unknown and complex spatial distributions, we reasoned that the localization precision offers an intuitive scale to cluster binding events, as localizations produced by a single isolated docking strand are

expected to reside within a circular area with radius of approximately twice the localization precision, while more complex shapes can be described by a superposition of those localization-precision-sized clusters.

To develop the clustering method, we imaged the commercially available GATTA-PAINT nanoruler, which has DNA-binding sites arranged linearly in 3 marks, with a mark-to-mark spacing of 80 nm (Fig. 2a). We then clustered the localizations using the DBSCAN algorithm[14], which has been widely used on SMLM data[15]. DBSCAN is a density-based algorithm that makes no assumptions about the number or shape of clusters but requires a search radius ($\varepsilon$) and a minimum number of localization ($n$) that are expected to reside within this search radius to form a cluster. Localizations that do not meet the above criteria are classified as noise. Given that a minimum of 5 binding events is required to run the statistical test, we set $n = 5$ and $\varepsilon$ to the median localization precision. Under these conditions DBSCAN sometimes identified and correctly assigned localizations from a single mark to a single cluster, but in most cases because of the partial overlap between the localizations from neighbouring marks, DBSCAN merged localizations from two, and in certain cases all three neighbouring marks, into a single cluster (Fig. 2b).

To break these larger clusters into smaller, localization-precision-sized, clusters we applied a second clustering step based on *k*-means clustering[16], which partitions the merged localizations into a predefined number of localization centroids. The number and start positions of the centroids were established by cross-correlating a 2D Gaussian template image with the reconstructed image of each cluster formed in the DBSCAN step (Supplementary Fig. 1) where the width of the 2D Gaussian template is equal to the localization precision. This second clustering step results in the expected number of 3 clusters per nano-ruler regardless of the initial DBSCAN clustering results (Fig. 2b).

Importantly, this two-step clustering method works well for more complex structures and groups localizations into localization-precision-sized clusters in both low- and high- density areas. To demonstrate this, we imaged E-cadherin-GFP in the follicle epithelium of stage 7 *Drosophila* egg chambers (Fig. 2c)[17]. E-cadherin is a homophilic transmembrane adhesion protein that plays a key role in the formation of the adherens junctions at the top of the lateral borders between epithelial cells. Cis interactions between E-cadherin molecules in the same cell lead to E-cadherin clustering, while adhesive

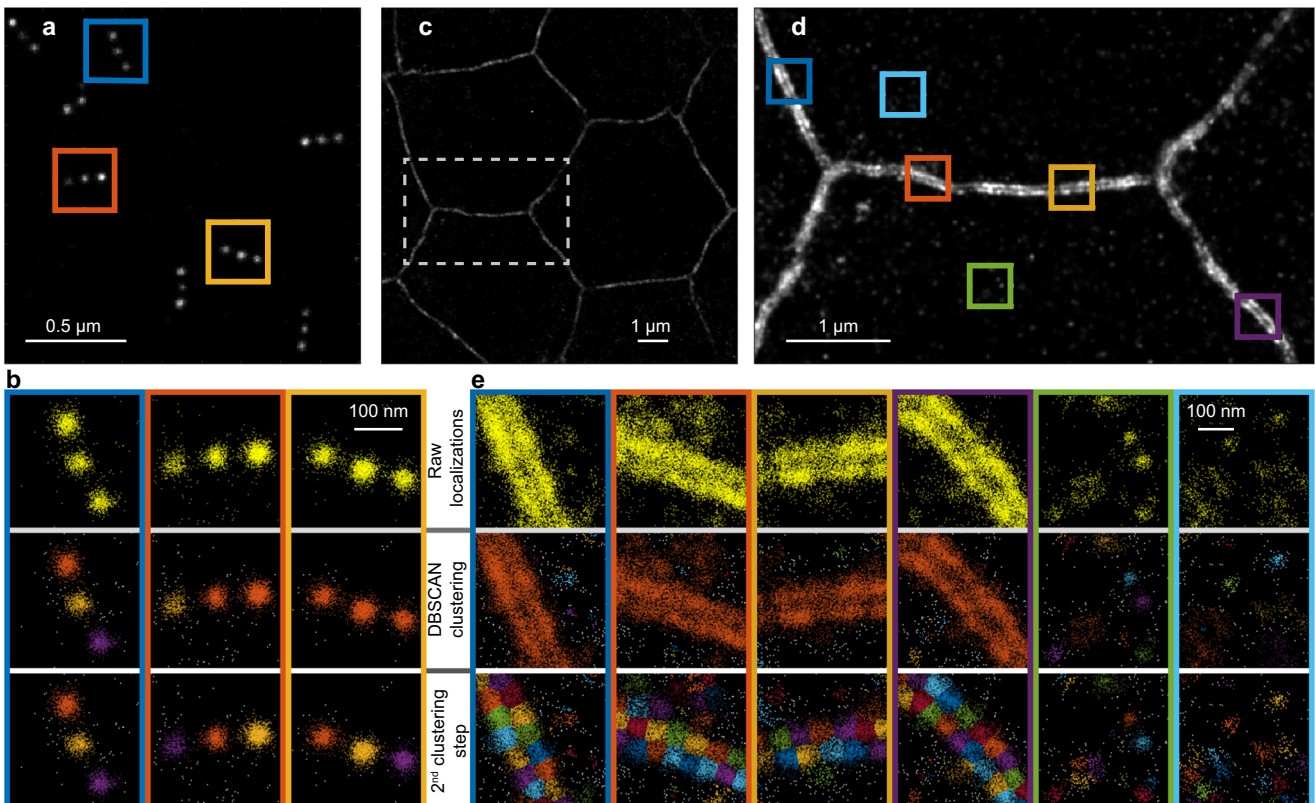

**Fig. 2 | Two step clustering method for generating localization-precision-sized clusters. a** SMLM image of the 80 nm GATTA-PAINT nanoruler. Images are representative of 4 independent experiments. **b** Examples of the nanoruler clusters marked by boxes in (**a**) before clustering (top), after DBSCAN clustering (middle) and after a second k-means clustering step (bottom). Clusters from the first DBSCAN clustering step can contain localizations from one, two or all three marks of the nanoruler. Localizations that have been classified as noise are shown in grey. The second clustering step partitions localizations from single marks into individual clusters. **c** SMLM image of E-cadherin, highlighting the borders between follicle cells in a *Drosophila* stage 7 egg chamber. **d** Image of the borders between

neighbouring cells, showing the clustered distribution of E-cadherin in the two cell membranes, which are resolved. Images are representative of 5 independent experiments. **e** Steps in the clustering of the binding events shown in the boxed regions in (**d**). In high-density regions near the cell borders, localizations from neighbouring cells are grouped together into large, elongated clusters during the first DBSCAN clustering step. The second clustering steps partitions these large clusters into smaller localization-precision-sized clusters. In low-density regions most of the localizations are clustered into localization-precision-sized clusters during the first clustering step and the second clustering step affects only the larger clusters.

trans interactions with E-cadherin in the neighbouring cell result in the alignment of these clusters in adjacent cell membranes[18]. This behaviour is captured in DNA-PAINT images, where juxtaposed traces of E-cadherin can be resolved in the membranes on both sides of a cell-cell junction (Fig. 2d). Because of the high density of localizations in this region, DBSCAN tends to cluster localizations into few elongated clusters, that can span both cell membranes (Fig. 2e). The second clustering step breaks these large clusters into smaller localization-precision-sized clusters that are restricted to one cell membrane (Fig. 2e).

E-cadherin is also present in the cytosol e.g. in endocytic and exocytic vesicles, but at a lower concentration than the lateral membrane. Here, most localizations are clustered into localization-precision-sized clusters during the first DBSCAN clustering step, and as expected, the second clustering step only partitions the larger clusters (Fig. 2e).

## Time trace construction

The next step uses the localizations in each cluster to construct a time trace of the binding events for that cluster. In DNA-PAINT, the binding time is determined by the homology between the imager and docking strand sequences and does not depend on the photophysics of the fluorophore attached to the imager strand. The duration of a binding event is inferred from the duration of a blinking event, with persistent blinks over consecutive frames counting as a single binding event and

blinks separated by one or more frames as different binding events (Fig. 3a). Because of fluorophore photophysics, however, a fluorophore can undergo several blinking events during the time its imager strand is bound to the docking strand, especially when imaging further away from the coverslip, which often entails the use of higher excitation power densities[19–21] than TIRF, for example. It is therefore important to account and correct for this photophysical blinking when constructing the time traces (Fig. 3a).

To assess the impact of photophysical blinking on time trace construction, we imaged the GATTA-PAINT nanoruler at a frame rate of 25 fps and constructed time traces using the localizations from individual marks, which, according to the manufacturer, are produced by binding events with an approximate range of ~0.5–1 s. If we only merge blinks that persist over consecutive camera frames, the cumulative distribution function (CDF) of the times between binding events poorly follows the expected exponential distribution (Fig. 3b). However, if we merge blinks that are separated by only a few dark frames, the RMSE error between the experimental CDF and an exponential model is reduced, indicating that the times between binding events follow the expected exponential distribution more closely (Fig. 3b). The similarity between the experimental CDF and the exponential model continues to improve as the time gap allowed between merged blinking events increases to ~15-frames (0.6 s) (Fig. 3c). This time coincides with the expected binding time, suggesting that to minimize the effect of photophysical blinking, it is important to merge blinks

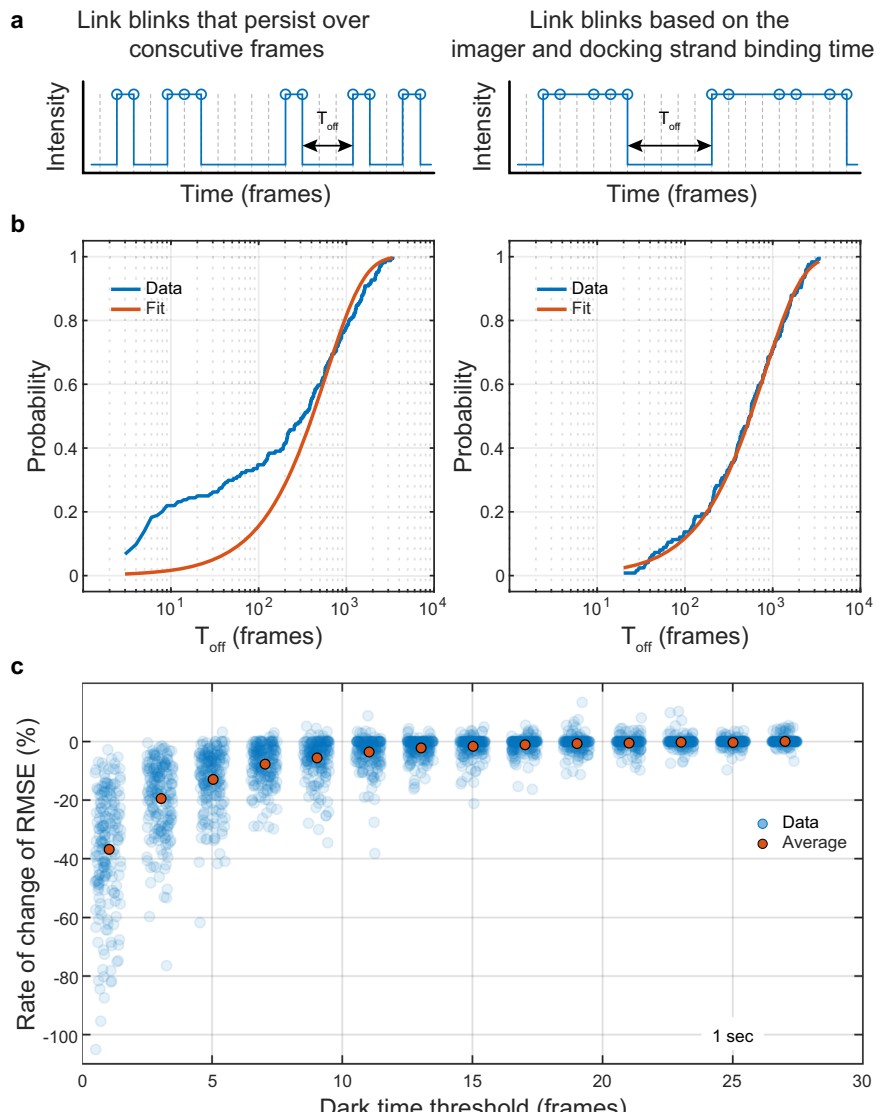

**Fig. 3 | Construction of time traces to measure the binding kinetics between the imager and docking strands. a** A time trace that represents binding events between the docking and imager strands can be constructed by linking blinks that persist over consecutive frames. Alternatively, we can allow a certain number of dark frames between blinks and construct a time trace by linking blinks that are separated in time by up to a dark time threshold set by the expected binding time between the imager and docking strands. **b** Measurements of the binding times from time traces constructed by only linking persistent blinks are susceptible to errors and result in a poor fit between the experimental CDF and an exponential distribution (lefthand graph). Time traces with blinks linked based on the expected binding time between the imager and docking strands provide more accurate measurements of the times between binding events and the experimental CDF follows more closely the expected exponential distribution (righthand graph). **c** The rate of change of the RMSE error between the experimental CDF and the expected exponential distribution as a function of the dark time threshold used to link the blinks.

that are separated by up to approximately the expected binding time. Allowing for longer times between blinks does not offer any further improvements and in extreme cases with high binding frequencies may erroneously lead to merging of consecutive, independent binding events. We note that the binding time may also be affected by the local environment of the docking strands, as we have observed longer binding times with docking strands conjugated to the HaloTag, where the oligonucleotide lies in a pocket containing the Halo active site (Supplementary Fig. 2).

## Test for exponentiality
**Determining the optimal parameters for the statistical test.** The last step in the denoising pipeline is to apply a null hypothesis test for exponentiality. We used the Anderson–Darling test to test for the null hypothesis that the times between binding events are drawn from an exponential distribution. The alternative hypothesis is that the times

between binding events do not follow an exponential distribution. The significance level of the test indicates the probability of erroneously assigning localization clusters produced by real hybridization events between imager and docking strands as non-specific signal. Common choices for the significance level are 5% or 1% and to determine which one is more appropriate in this case, we used the GATTA-PAINT nanoruler. We selected regions with the characteristic 3 DNA-binding marks (Fig. 4a) and performed the 2-step clustering method described above to separate localizations into 3 clusters, analysing a total of 555 individual clusters. For each cluster, we linked the blinks based on the expected binding time between the imager and docking strands, allowing for a gap of up to 1 s, to obtain the times between binding events (Fig. 4b). We then applied the Anderson–Darling test to each individual cluster and plotted the fraction of non-rejected clusters as a function of significance level (Fig. 4c). Within the investigated range, the fraction of non-rejected clusters exhibited a near-monotonic

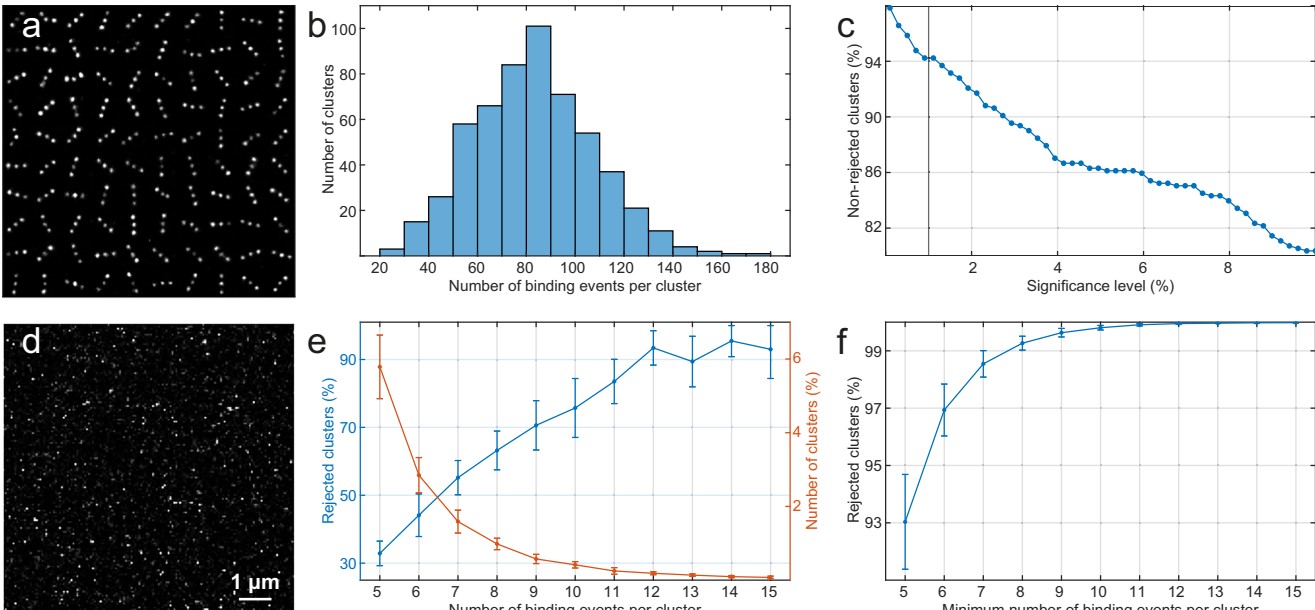

**Fig. 4 | Experimental determination of the optimal parameters for the Anderson–Darling null hypothesis test. a** Montage image of the GATTA-PAINT nanorulers exhibiting the characteristic linear arrangement of DNA-binding sites in 3 marks, with a mark-to-mark spacing of 80 nm. **b** A histogram showing the distribution of the number of binding events per individual cluster of the GATTA-PAINT nanorulers. **c** The fraction of non-rejected clusters as a function of the significance level for the Anderson–Darling test. **d** A super-resolved image of an E-cadherin negative control sample that has not been exposed to docking strands. Image is representative of 7 independent experiments. **e** The percentage of clusters with a given number of binding events (orange line) and the percentage of these clusters that are rejected by the Anderson–Darling test (blue line). **f** Total percentage of non-specific clusters rejected as a function of the minimum number of binding events required per cluster. Clusters with fewer binding events are directly rejected, while the Anderson–Darling test is applied to clusters with more binding events. Error bars indicate mean ± standard deviation (E-cadherin negative control samples $n = 7$).

decrease with increasing significance level, suggesting that a 1% significance level is more appropriate, as this level leads to the acceptance of more than 94% of the real clusters.

To determine the power of the test to correctly reject background clusters with non-exponential binding kinetics, we imaged follicle cells in *Drosophila* egg chambers expressing E-cadherin-GFP that had not been labelled with the docking strand-coupled GFP-nanobody but were otherwise prepared and imaged under standard conditions. In these samples, all the observed blinks are due to non-specific interactions between the fluorophore-coupled imager strand oligonucleotides and the sample (Fig. 4d). The binding time for these non-specific interactions is not known, but to maintain consistency in our analysis, we linked the non-specific blinks based on the expected binding time between the docking and imager strands to obtain the number of apparent binding events per cluster and the times between these apparent binding events. We then separated the clusters in groups based on the number of binding events per cluster, ran the Anderson–Darling test for each group and measured the fraction of rejected clusters. By plotting the fraction of rejected clusters versus the number of binding events per cluster, we observed that ~33% of the clusters with only 5 binding events were rejected, but the fraction of rejected clusters increased with increasing number of binding events per cluster and reached a plateau of >90% for clusters with ≥12 binding events, indicating that the power of the test increases with increasing sample size (binding events) up until this point and then remains constant (Fig. 4e). Since a large proportion of the background clusters contain few binding events, introducing a threshold on the minimum number of binding events per cluster provides an effective way to reject non-specific clusters that have too few blinks to apply the Anderson–Darling test effectively. In this way, clusters with fewer binding events than the threshold value, where the power of the test is low, will be directly rejected, while non-specific clusters with more binding events than the threshold value will be rejected because of the

increasing power of the test. This cumulative effect is shown in (Fig. 4f) where the fraction of rejected clusters is plotted as a function of the minimum number of binding events per cluster. Here >95% of the non-specific clusters are rejected with a threshold of 6 binding events per cluster while the fraction of rejected clusters rises to >99% for 8 binding events per cluster.

The optimum value for the threshold of minimum binding events per cluster depends on the biological application. We suggest using a minimum of $N = 12$ binding events per cluster for quantitative methods such as qPAINT[2], since this is the point at which the Anderson–Darling test becomes optimally effective. qPAINT counts the number of molecules present in a cluster by measuring the times separating consecutive binding events to estimate the mean time between binding events. The error on this estimate scales as $\frac{1}{\sqrt{N-1}}$, where N is the number of binding events in the cluster. Thus, applications of qPAINT that collect enough data to reduce this error will usually have values of N that are higher than 12.

For other biological applications the optimum threshold value can be obtained from the super-resolved images themselves by comparing background regions with regions where most signal is specific. Here, setting the threshold involves a balance between increasing the efficiency of the method in rejecting non-specific clusters and minimizing the number of specific clusters that are rejected because their number of binding events fall below the threshold.

**Demonstration of the denoising method.** To demonstrate this approach, we imaged E-cadherin in follicle cells in stage 7 *Drosophila* egg chambers, but this time we induced mitotic recombination to generate clones of cells that lack E-cadherin-GFP next to cells that are homozygous for E-cadherin-GFP (Fig. 5a). The juxtaposition of E-cadherin-GFP expressing cells with non-expressing cells offers a straight-forward approach to evaluate the effectiveness of our method

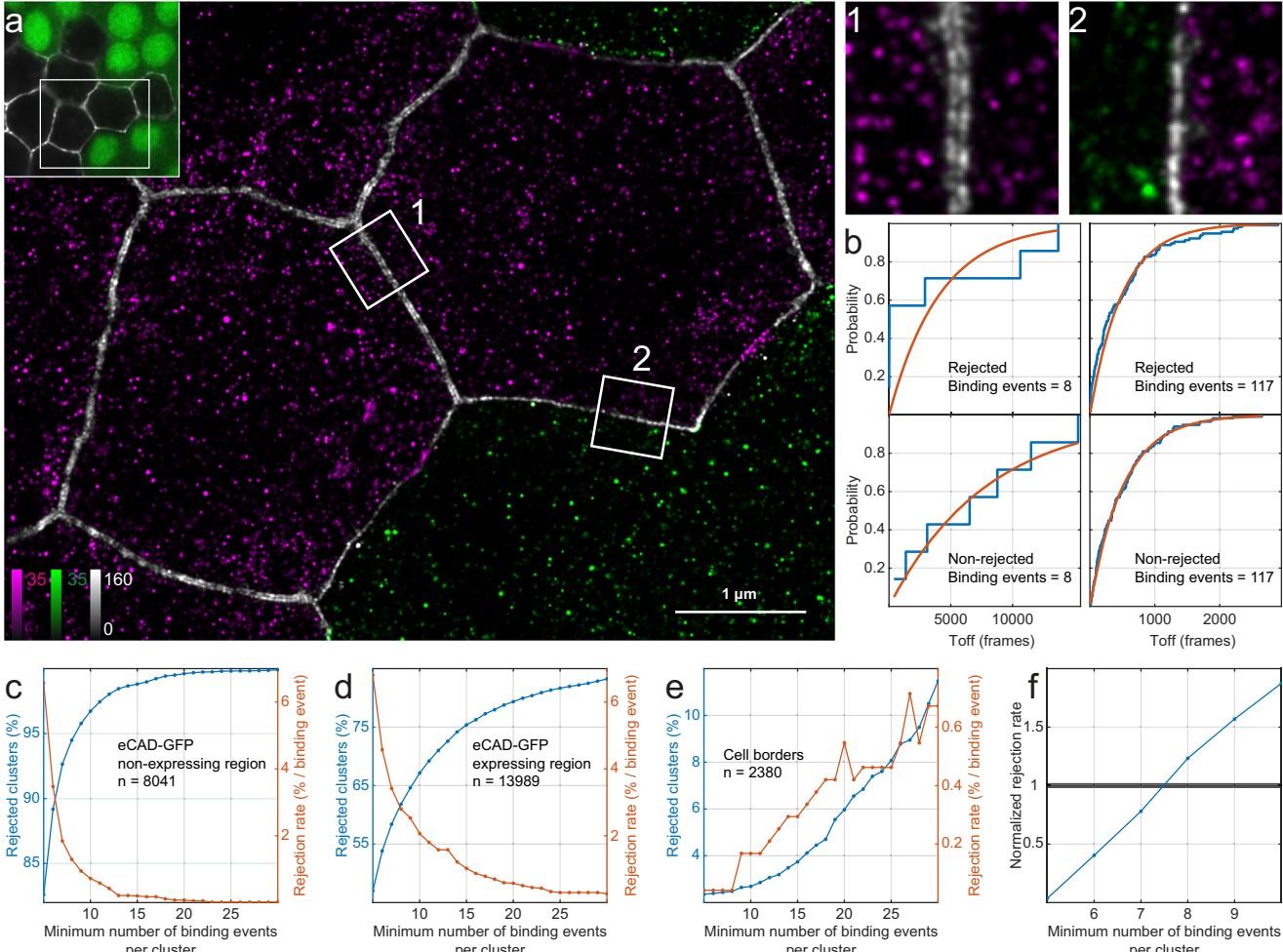

**Fig. 5 | Demonstration of the denoising method. a** Overview image of a mosaic epithelial tissue of stage 7 *Drosophila* egg chamber containing cells that lack E-cadherin GFP next to cells that are homozygous for E-cadherin GFP. Non-GFP cells are marked by a visible nucleus shown in green. Image is representative of 3 independent experiments. In the super-resolved image epithelial cells homozygous for E-cadherin GFP are shown in magenta next to non-GFP cells shown in green. Borders between cells are illustrated in grayscale. Borders between cells homozygous for E-cadherin-GFP appear as double lines (inset 1) while borders between homozygous E-cadherin-GFP cells and non-expressing cells appear as single lines (inset 2). Borders between non-expressing cells are not visible. **b** Example cumulative distributions of the times between binding events along with an exponential fit (shown in red) for non-rejected and rejected clusters with low and high numbers of binding events. **c–e** The fraction of rejected clusters (blue line; total percentage of clusters rejected as the minimum number of binding events required per cluster increases) and cluster rejection rate (orange line: the change in the percentage of clusters that are rejected when the minimum number of binding events is increased by one) for the non-expressing region, the expressing region, and the cell borders respectively. Cluster rejection rates were smoothed using a 5-point moving box median. **f** Normalized rejection rate of the expressing to the non-expressing region as a function of the minimum number of binding events per cluster. The data were smoothed using a 3-point moving box averaging. The optimal threshold of minimum number of binding events for the denoising method is determined when the normalized rejection rate exceeds 1.

in removing non-specific localizations, as both the sample and the control undergo identical sample preparation steps and are imaged under identical conditions. In the super-resolved images, almost all clusters that lie in the adherens junctions along the borders between E-cadherin-GFP expressing cells represent specific interactions between the imager and docking strands, whereas the non-expressing clones are regions where non-specific interactions prevail. Note that borders between E-cadherin-GFP expressing cells appear as double lines, because the E-cadherin clusters in the two opposing cell membranes are resolved, whereas borders between E-cadherin-GFP expressing cells and non-expressing cells appear as a single line (Fig. 5a 1-2). In the non-expressing region, the borders between cells are not visible. We then defined the clusters in these regions and analysed whether the temporal distribution of blinks in each cluster passed the Anderson–Darling test for exponentiality (Fig. 5b). Counting the clusters in each region gives the fraction of rejected clusters as a function of the minimum number of binding events as shown in Fig. 5c–e.

The E-cadherin-GFP non-expressing region is dominated by non-specific signal and largely behaves like the negative control sample. However, only ~89% of clusters are rejected with a threshold of 6 binding events compared to >95% in the negative control (Fig. 5c). These extra accepted clusters presumably correspond to signal from docking strands coupled to off-target nanobodies. The signal from these docking strands is indistinguishable from real signal and will follow the expected DNA-binding kinetics. The corresponding clusters will therefore not be rejected by the Anderson–Darling test.

The E-cadherin-GFP expressing region contains both specific and non-specific clusters. The rejection rate ($r_s$) in this mixed region will therefore decrease more slowly than in the non-expressing region ($r_b$) as the minimum binding event threshold is increased, although they both start at approximately the same level (Fig. 5d). This can be explained by the prevalence of specific clusters in the expressing region. Non-junctional E-cadherin is found in endocytic and exocytic vesicles, and recycling endosomes, with each compartment carrying

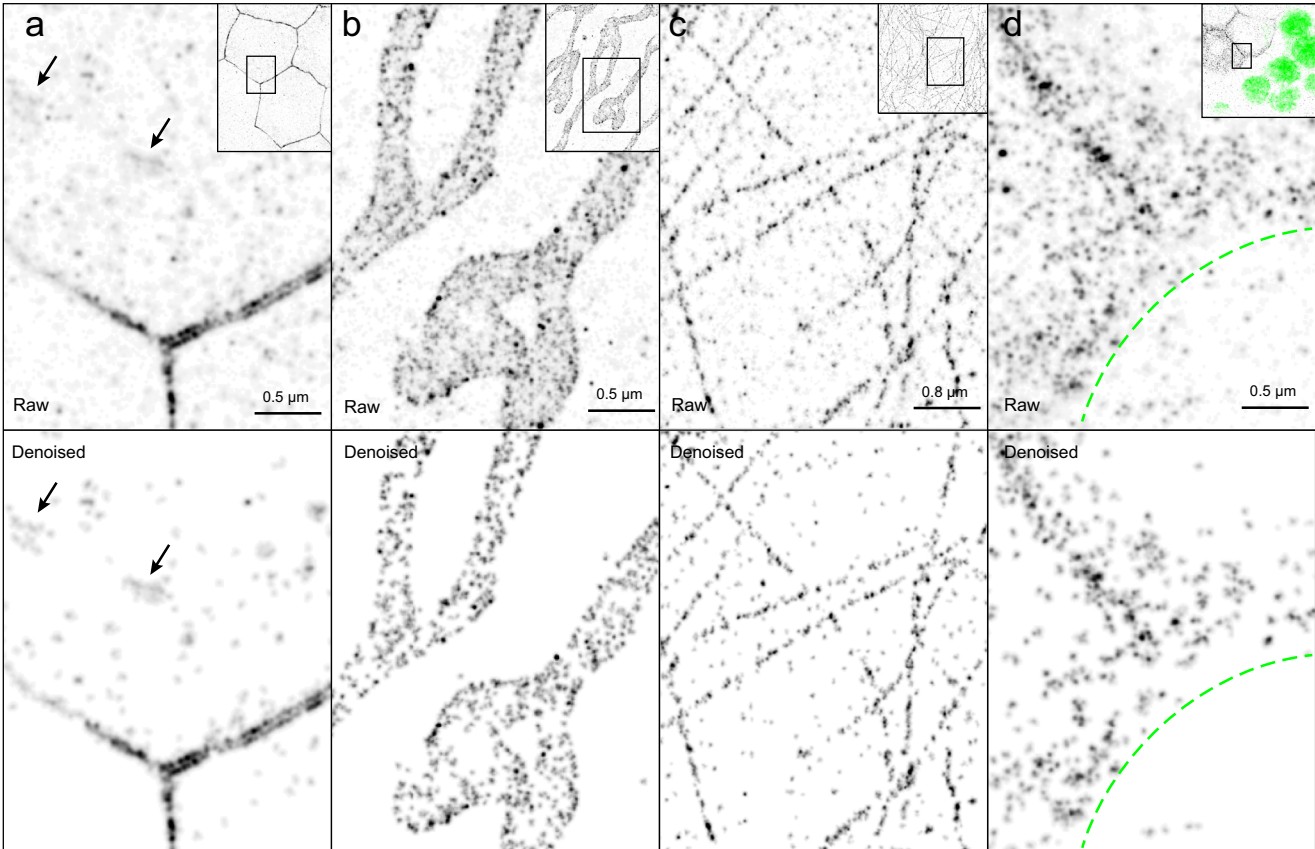

**Fig. 6 | Comparison between raw and denoised DNA-PAINT images. a** E-cadherin in mosaic epithelial tissue of a stage 7 *Drosophila* egg chamber. Images are representative of 3 independent experiments. E-cadherin highlights the cell-borders and is also present in endocytic and exocytic vesicles in the cytosol. Arrows point to tubular E-cadherin structures, most likely recycling endosomes, that emerge with sharper outlines after denoising. **b** mitochondria in U2OS cells. Images are representative of 5 independent experiments. The characteristic shape of mitochondria offers a clear distinction between areas in the sample where specific and non-specific signal is expected. The areas of the sample with non-specific signal should be used with caution as a reference for denoising the areas with specific signal, as off-target docking strands can result in excessive removal of specific signal. **c** Microtubules in HeLa cells. Images are representative of 4 independent experiments. Additional DNA-PAINT images of HeLa cells that were not exposed to docking strands, but were otherwise prepared and imaged under identical conditions, serve as the negative control required by the denoising method. **d** aPKC in mosaic epithelial tissue of a stage 7 *Drosophila* egg chamber. Images are representative of 3 independent experiments. The inset shows an overview image with cells expressing Halo-aPKC next to cells not expressing Halo-aPKC which are marked by a visible nucleus shown in green. In the super-resolved image a green dashed line serves as a guide-to-the-eye for the boundary between those expressing and non-expressing cells. aPKC is enriched towards the apical side of the cells but it does not assume a characteristic shape and does not provide visual cues to aid in the separation of specific from non-specific clusters. This crucial step is performed by the denoising method.

multiple molecules of E-cadherin. The specific clusters in this region will therefore contain many E-cadherin molecules and exhibit a higher number of binding events. These clusters are not eliminated by the test but are gradually rejected for failing to meet the threshold for the minimum number of binding events per cluster as the threshold increases. E-cadherin is highly enriched in the junctional region where both the concentration of specific clusters and the number of docking strands per cluster is expected to be the highest. Here rejection levels remain below 4% when the minimum number of binding events per cluster is 12 (Fig. 5e).

The background, non-specific clusters are presumably present at equivalent levels in both the expressing and non-expressing regions, but the E-cadherin specific clusters occur only in the expressing region. The percentage of clusters that are rejected should therefore be lower in the expressing region for low threshold values. As the threshold for the minimum number of binding events per cluster increases and the non-specific clusters are depleted from both regions, the rejection rate in the expressing region will become higher than the non-expressing region because some specific clusters are erroneously rejected from the remaining population. Thus, the optimum threshold value can be calculated by determining when the rejection rate in the non-

expressing region ($r_b$) equals the rejection rate in the expressing region ($r_s$). This value can be determined graphically by normalizing the rejection rate in the expressing region to the one from the non-expressing region ($\frac{r_s - r_b}{r_b}$) and plotting it as a function of the minimum number of binding events per cluster (Fig. 5f). In this case we can set the threshold to the minimum number of binding events where the normalized rate exceeds one, which in this case is 8 binding events. Under these conditions we reject >90% of the clusters in the non-expressing region, whereas <2% from the signal at the cell border is rejected. The denoised image is shown in Fig. 6a. It should be noted that in contrast to the cell-cell junctions, where E-Cadherin is enriched, identifying E-cadherin clusters in vesicles in the cytoplasm and filtering them out from the non-specific signal would be very difficult otherwise as they cannot be morphologically identified. Additionally, tubular structures, probably recycling endosomes, become sharper and more prominent[22,23].

**Denoising using parts of the image that lack specific signal.** This method for setting the threshold of minimum number of binding events per cluster is not limited to mosaic tissues but is directly

applicable to all images where a clear distinction between areas with non-specific signal and areas with specific signal can be established. To demonstrate this, we imaged TOM20 labelled mitochondria in U2OS cells (Fig. 6b). The characteristic shape of mitochondria facilitates the segmentation of the super-resolved images into areas with and without specific signal (Supplementary Fig. 3) and a denoised image is readily obtained with ~94% of the clusters in areas of the image not occupied by mitochondria rejected as shown in (Fig. 6b). We note that areas of the image with non-specific signal should be used with caution to estimate the background levels, since all signals in the areas outside the mitochondria are considered as noise, and this includes signal from off-target binding of docking strands. This signal from off-target docking strands is indistinguishable from specific signal and may result in the removal of excessive amounts of specific signal during the denoising process (Supplementary Fig. 3).

**Denoising using a negative control.** To avoid this problem and minimize the loss of specific signal, a negative control that lacks docking-strands can be used instead. To demonstrate this approach, we imaged microtubules in HeLa cells transfected with GFP-tagged α-tubulin (Fig. 6c, Supplementary Fig. 5a). Images of these cells acquired under standard DNA-PAINT imaging conditions without labelling the GFP with a docking strand serve as the negative control. They are used to denoise DNA-PAINT images of microtubules in separately prepared transfected cells that were processed under identical conditions except that GFP was labelled with a single domain antibody coupled to a docking strand (Supplementary Fig. 5b). Here, the denoising method removes >97% of the clusters in the negative control and ~42% of the clusters in the labelled sample image, most of which lie in areas between the microtubules (Fig. 6c, Supplementary Fig. 5).

Using a negative control sample to characterize the non-specific signal required for the denoising method is a more general method and can be applied in all cases but is more labour intensive and susceptible to experimental variations. Therefore, it is important to ensure that the negative control is prepared and imaged under standard experimental conditions.

**Denoising samples with amorphous protein distributions.** A key feature of our method is that it does not make any assumptions on the underlying distribution of the protein of interest, hence is directly applicable to proteins with amorphous distributions. To demonstrate this, we imaged the follicle cell epithelium in stage 7 *Drosophila* egg chambers in which clones of cells homozygous for Halo-tagged atypical protein kinase C (Halo-aPKC) lie adjacent to cells that do not express Halo-aPKC (Fig. 6d). aPKC is a cytosolic protein that is enriched at the apical side of epithelial cells, but it does not assume a characteristic shape. Here, the non-expressing region is obtained from a clone of Halo-aPKC non-expressing cells, similar to the approach taken with E-cadherin (Fig. 6d Inset, Supplementary Fig. 4). By clustering and linking blinks with a 2 sec gap (Supplementary Fig. 2), the denoising method eliminates ~96% of the clusters in this region with a threshold for the minimum number of binding events of 8. We note that non-specific clusters are indistinguishable from real aPKC clusters in the super-resolved images due to the amorphous distribution of aPKC. Denoising is therefore essential for obtaining accurate images of the spatial arrangement of aPKC (Fig. 6d).

## Discussion

All light-based microscopy suffers from non-specific background signal that reduces the quality of the resulting images. However, this is a particular problem for any analyses of DNA-PAINT data aimed at determining spatial relationships between different proteins as non-specific clusters produced by imager strands for one protein can erroneously co-localise with the real signal for the other protein. Furthermore, the presence of non-specific clusters also compromises

measurements of the amounts of a protein using qPAINT. A key step in this process is to determine the labelling efficiency of the protein, which can be estimated from the colocalization of two tags attached to the same protein[24]. Without denoising, measurements of the labelling efficiency will be inaccurate because of excess unpaired non-specific signal and/or false colocalizations of each tag with non-specific signal. Denoising is therefore a crucial step that removes non-specific clusters for any downstream analysis and can be combined with existing quantitative methods to obtain more accurate measurements on spatial relationships and proteins quantities.

Here we introduce a denoising method that exploits the predictable kinetics between docking and imager strands and uses a statistical test to differentiate between non-specific and specific binding events in DNA-PAINT super-resolved images. Our analysis explicitly considers the binding time between the imager and docking strands, which is essential to minimize artifacts from photophysical blinking of fluorophores and obtain an accurate representation of the DNA-binding kinetics between the imager and docking strands. This correction for multiple blinks from a single binding event is also essential for counting the number of protein molecules using qPAINT.

We used the Anderson–Darling test with a null hypothesis that the times between DNA-binding events follows an exponential distribution, without making any assumptions on the alternative distribution for the non-specific binding events. We also considered using the duration of the binding events, instead of the times between DNA-binding events. Binding times are also expected to follow an exponential distribution, but with a very characteristic time determined only by the imager and docking strand sequences, which in our case is ~1 s. However, we did not pursue this approach as this would require measuring the duration of binding events with higher accuracy, and therefore imaging at higher frame rates. Given the limited photon budget under our experimental conditions, imaging at higher frame rates would negatively impact the final super-resolved images due to a lower number of detected binding events, and reduced localization precision.

The Anderson–Darling test requires a minimum of 5 binding events and for this number of binding events we experimentally determined the power of the test at ~33%. However, the power increases with increasing amounts of data to >90% for 12 binding events. To determine the optimal value for the minimum number of binding events from the super-resolved images, we propose a method that compares the cluster rejection rates between regions of the sample dominated by non-specific signal and regions with specific signal. It is important to note that all signal in the non-specific area of the sample will be considered as noise, and it is therefore crucial to minimize any contributions from off-target binding of docking strands by following best practices during sample preparation. Alternatively, one can use a negative control sample that has not been exposed to docking strands, but is otherwise prepared and imaged under identical conditions. This approach avoids the risk of removing too much specific signal and is a more general method for measuring the non-specific signal required for denoising, as it can be applied in cases where regions dominated by non-specific signal are not available in the sample. Finally, care must be taken to minimize docking strand photodamage during imaging to ensure that the number of binding events per docking strand is maximised, as this improves the power of the statistical test and increases the levels of specific signal. Under typical imaging conditions with less than 10% docking strand loss during imaging[7], simulations suggest that the denoising algorithm is insensitive to docking strand depletion. However, excessive docking strand photodamage may cause the apparent distribution of the times between binding events to deviate from exponential, resulting in the rejection of specific clusters.

For quantitative applications such as qPAINT that count the molecules present in a cluster by estimating the mean time between binding events from an exponential fit as shown in (Fig. 5b), we suggest

using a minimum of 12 binding events per cluster to increase confidence that the cluster is specific and improve the accuracy of the estimate of the mean dark time. Of course, a higher number of binding events per cluster would provide an even better estimate of the mean time between binding events but at a cost of longer acquisition times. Therefore, it is important to adjust the concentration of the imager strands and/or the acquisition times accordingly.

In summary, we showed that partitioning the data in localization-precision-sized clusters offers a straightforward way to probe the binding kinetics and identify specific clusters even in otherwise difficult situations where no additional information from their spatial distribution is available. The denoising method can be applied to, and improve, all DNA-PAINT images, but most importantly can be combined with existing methods aiming at determining absolute amounts and spatial relationships of proteins.

## Methods

### Microscope setup
Data collection was carried out on a custom-built spinning disk microscope equipped with a spiral disc from CrestOptics as previously reported[20]. All imaging was carried out with a 100 × 1.35NA silicone immersion objective lens (Olympus, UPLSAPO100XS). Fluorescence is collected with this objective lens and relayed to the primary image plane at the side port of the microscope base where the spiral disk is positioned. After passing through the spiral disk, fluorescence is separated from excitation laser light with a dichroic mirror (Chroma, ZT543rdc or ZT405/488/561/647rpc) and an additional bandpass filter (Chroma, ET560lp or ZET405/488/561/647). The primary image plane is relayed to an intermediate plane, using a 1X telescope, where an adjustable rectangular aperture (Owis, 27.160.1212) is positioned. This intermediate image plane is subsequently relayed to the camera sensor plane and passes through an additional bandpass filter (Chroma, 569/25×). A sCMOS (Hamamatsu, Orca Flash 4.0V3) was used for image collection with an effective pixel size of ~98 nm. Microscope control and data collection was carried out on a standard Windows based PC running a custom microscope control program developed in the LabView environment and available on GitHub (https://github.com/Gurdon-Super-Res-Lab/Microscope-Control).

### *Drosophila melanogaster* lines
**Standard procedures were used for *Drosophila* maintenance and experimentation.** *D*E-Cadherin images were taken using the GFP knock-in lines of Huang et al.[17]. *D*E-Cadherin::GFP clone images were obtained by recombining *D*E-Cadherin::GFP with FRT-G13 (w[*]; P{w[+mW.hs] = FRT(w[hs])}G13 BL1956), and then crossing to hs-FLP; FRT-G13, Ubi-mRFP.nls/CyO flies to obtain hs-FLP/+; FRT-G13, *D*E-Cadherin::GFP/FRT-G13, Ubi-mRFP.nls progeny. Clones were obtained by heat-shocking either pupal or adult stages for 3–4 days at 37 °C for 2 h 2× per day. Flies were then fattened on yeast for 1–2 days at 25 °C before dissection.

aPKC images were taken using the Halo-SNAP (HaSP)-aPKC CRISPR line from Erdmann et al.[25]. Halo-aPKC clone images were obtained by recombining HaSP-aPKC with FRT-G13 (w[*]; P{w[+mW.hs] = FRT(w[hs])}G13 BL1956), and then crossing to hs-FLP; FRT-G13, Ubi-mRFP.nls/CyO flies to obtain hs-FLP/+; FRT-G13, HaSP-aPKC/FRT-G13, Ubi-mRFP.nls progeny. Clones were obtained by heat-shocking either pupal or adult stages for 3–4 days at 37 °C for 2 h 2× per day. Flies were then fattened on yeast for 1–2 days at 25 °C before dissection.

Nup160 images were taken using the Nup160-Halo CRISPR line from Cheng et al.[21]. Freshly eclosed flies were fattened on yeast for 1–2 days at 25 °C before dissection.

***Drosophila* sample preparation.** Fattened ovaries were dissected in Schneider's medium supplemented with insulin (7.5 μg/ml, Merck I9278). Ovarioles were dissected out of the surrounding muscle sheath

and were fixed for 20 min at room temperature with prewarmed (38 °C) fixative (4% methanol-free formaldehyde in 0.5× PBS). Samples were quenched by rinsing and then washing for 5 min in 50 mM ammonium chloride. They were then permeabilised in 0.2% Triton X-100 in PBS for 5 min, followed by 3× 5 min washes in PBS. Labelling was performed for 1 hour at 37 °C with shaking in 0.5% bovine serum albumin (BSA), 1 μM DTT in PBS with either a 1:300 dilution of the GFP-nanobody conjugated to docking strand (Massive Photonics, Massive Tag-Q Anti-GFP Kit) for GFP labelling or 1 μM Halo-ligand conjugated to the P3 docking strand for Halo labelling[3]. After labelling, samples were washed 3 × 5 min with PBS and then overnight with PBS + 0.1% Triton X-100. The next day, samples were washed 3× 20 min with PBS and mounted on 8-well chambered coverslips (Ibidi 80807) that had been treated with Cell-Tak (Corning 354240). Samples were imaged in 300 μl of imaging solution (PBS + 500 mM NaCl + 20 mM sodium sulphite + 1 mM Trolox) with 2 nM Imager 3 Cy3B (Massive Photonics) for GFP, or 2 nM P3 Cy3B imager strand for Halo.

### Mitochondria sample
We utilized the Massive-PAINT 2-PLEX kit from Massive Photonics for mitochondria imaging in U2OS cells which includes all probes and reagents.

### Cell culture, transfection, fixation and staining for microtubule sample
HeLa Kyoto cells were obtained from the Research Institute of Molecular Pathology (Vienna, Austria). Cells were cultured in media composed of DMEM GlutaMAX (Gibco, Thermo Fisher Scientific) supplemented with 10% foetal bovine serum (FBS), 1% penicillin-streptomycin and 1% L-Glutamine (all from Gibco, Thermo Fisher Scientific) at 37 °C and 5% CO$_2$ and were regularly tested for mycoplasma contamination.

Cells were plated onto 8-well glass bottom chambered coverslips (Ibidi, 80807) 24 h prior to transfection with a pEGFP-Tub plasmid (Clontech/Takara, 632349) to label α-tubulin. For each well, 0.25 μg DNA with 0.75 μL FUGENE HD Transfection reagent (Promega, E2311) in 125 μL Opti-MEM reduced serum medium (Gibco, Life Technologies) were used according to manufacturer's instructions. Transfected cells were incubated in culture conditions as above for 36 h before fixation.

Cells were fixed and permeabilized simultaneously in 4% methanol-free formaldehyde (FA, Fisher Scientific) with 0.2% Triton in Cytoskeleton Buffer with Sucrose (CSB + S, 10 mM MES, 138 mM KCl, 3 mM MgCl2, 2 mM EGTA, supplemented with 4.5% sucrose w/v, pH 7.4) for 6 min at 37 °C, following which they were fixed in 4% methanol-free formaldehyde in CSB + S for 14 min at 37 °C and washed in PBS. Prior to staining, fixed cells were re-permeabilised in 1% Triton in PBS for 5 min at room temperature, blocked in Antibody incubation buffer (Massive Photonics) for 30 min, and incubated with 1:300 anti-GFP sdAB in the same Antibody incubation buffer for 45 min before being washed in Washing buffer (Massive Photonics).

### SMLM data acquisition and analysis
All SMLM data was acquired on the microscope described above. For each acquisition, unprocessed camera frames were acquired as sixty 1000 frame blocks (cycles) and a camera exposure time of 200 ms (5 Hz). Drift correction was performed during imaging by measuring the XYZ position of the sample at a rate of ~0.5 Hz using a near IR transmitted light lamp and camera separate from the primary imaging path. Full details of this scheme have been reported previously[21]. Unprocessed camera frames were analysed using custom localization scripts developed in the MatLab environment and using the previously reported analysis/fitting package from Li and co-workers[26]. Additional XY drift correction was carried out in postprocessing using the previously reported redundant cross correlation algorithm from Wang and co-workers[27].

## Denoising analysis

Raw localizations were first clustered using the DBSCAN algorithm with search radius $\varepsilon$ = the median localization precision, and a minimum number of localizations that are expected to reside within this search radius $n = 5$. Subsequently a super-resolved image was reconstructed for each DBSCAN cluster and cross-correlated with a 2D gaussian template with a sigma value set by the localization precision to determine the number and initial positions of centroids for a second k-means clustering step, which further partitions the localizations into localization precision-sized clusters. Typical processing times for a dataset with approximately two million (x, y) coordinates was 40 min for completion of both clustering steps on a standard PC running 64-bit Ubuntu with 32 GB RAM and an Intel i7 CPU at 3.2 GHz (example processing times and computer specifications are included with the example MATLAB scripts in the supplement). Localizations in each cluster separated by <2 s for Halo-tagged target and <1 s for all other tags were further linked to a single binding event to construct a time trace of the binding activity for each cluster and measure the times between the binding events. Finally, the distribution of the measured times between binding events for each cluster was tested against an exponential distribution with the Anderson–Darling test. Test results with a $p$-value < 0.01 were rejected as noise.

## GATTA-PAINT nanoruler imaging

Data using the GATTA-PAINT nanoruler (GATTAquant HiRes 80R) were collected using the 70 μm spirals and 642 nm excitation with a power density of 5 kW/cm$^2$. Camera frames were acquired in blocks of 1000 frames at a rate of 25 frames per second. In total 100,000 frames were collected for each image. Drift correction and localization analysis were performed as described for the SMLM data.

## Docking strand depletion simulations

To estimate the effect of docking strand depletion on the performance of the denoising method, we ran simulations on 5000 clusters with 10 docking strands per cluster, assuming an average binding time of 1 s, an average time between binding events for a single docking strand of 1000 s, and a total acquisition time of 10,000 s (-1.3 h). We investigated a typical case of docking strand depletion with 10% of the docking strands depleted during imaging and an average lifetime of a docking strand of 80,000 s, and a more extreme case with -60% of the docking strand depleted during imaging and an average lifetime of a docking strand of 10,000 s. Under the typical scenario the denoising method rejected -1% of the clusters, while under the more extreme case the denoising method rejected 6% of the clusters.

## Reporting summary

Further information on research design is available in the Nature Portfolio Reporting Summary linked to this article.

## Data availability

Localization data of all DNA-PAINT images in this manuscript are available on Zenodo (https://doi.org/10.5281/zenodo.18213668). Source data are provided with this paper.

## Code availability

The MATLAB scripts used for each step of the denoising method with example data may be freely accessed on GitHub (https://github.com/Gurdon-Super-Res-Lab/DNApaintDeNoise.git).

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

## Acknowledgements

This work was supported by a Wellcome Principal Fellowship to DStJ (224402/Z/21/Z), a BBSRC project grant (BB/V008595/1) and by centre grant support from the Wellcome Trust (24843). J.L.-L. is an Oliver Gatty PhD student.

## Author contributions

D.St J., G.S. and E.S.A. conceived the project. G.S. and E.S.A. developed the denoising algorithm. G.S. and E.S.A. analysed all the data. G.S. performed the imaging experiments with the GATTA-PAINT nanoruler. J.H.R. made the fly lines, prepared and imaged the *Drosophila* samples. J.L.-L. and E.K.P. provided help with the HeLa cells and E.S.A. performed the imaging experiments. D.StJ. acquired funding and supervised the project. G.S. and D.StJ. wrote the manuscript with input from all authors. All authors participated in the discussions and data interpretation.

## Competing interests

G.S., E.S.A. and D.StJ. filed a patent application covering the concept behind this study. The remaining authors declare no competing interests.
