## [Transparent Peer Review file · Nature Communications]

Utilizing the predictable binding kinetics of DNA-PAINT to denoise super-resolution images

Corresponding Author: Dr George Sirinakis

Version 0:

Reviewer comments:

Reviewer #1

(Remarks to the Author)

The manuscript entitled "Utilizing the predictable kinetic of DNA-PAINT to denoise super-resolution images" describes a process to 'denoise' DNA-PAINT data. DNA-PAINT uses 'imagers', fluorophores attached to diffusing oligonucleotides, to produce stochastic single molecule events. The predictable binding behavior of these small oligonucleotides has enabled the authors to describe a method to remove localization data suspected of originating from non-specific event detections. The authors have demonstrated the procedure on both synthetic DNA-origami as well as biological, *Drosophila* & cultured U2OS, samples. The manuscript could be further improved by addressing the following comments:

1. In the second paragraph of the introduction the authors talk about circumventing the problem of photobleaching. Whilst this is true the authors should introduce the concept of docking strand 'site-loss' caused by photodamage. This process can lead to a decrease in localizations over time similar to photobleaching. (see: <https://doi.org/10.3390/molecules23123165>)
2. Main Figure 1. The clustering step 1 and 2 boxes would benefit from additionally having "DBSCAN" and "k-means" mentioned within them. It would also be my preference to have greater detail in the figure legend.
3. Main Figure 4a. Scale bar and/or define the expected size of the nanoruler in the legend.
4. Main Figure 4d. The background/non-specific events you show here appears alarmingly high. Do you typically see higher levels of non-specific binding in your *Drosophila* samples in comparison to U2OS cells? Is 2 nM a typical imager concentration you would image with?
5. Additional media files (a hundred frames or so) that show the raw 'blinks' from binding/non-specific events in your various samples would be appreciated.
6. The authors should discuss the combination of imaging sparsely distributed super-resolution targets and how the effects of DNA-PAINT site-loss could affect their denoising approach.
7. The comparison between raw and denoised super-resolution images in Figure 6 highlights a particular concern that the denoising procedure could remove specific events. In Figure 6b, showing the mitochondria, it is suggestive that potentially specific events have been removed from within the mitochondrial signal leaving a number of voids. Can the authors alleviate this concern?
8. Have the authors tried fluorogenic DNA-PAINT (doi.org/10.1038/s41592-022-01464-9)? Perhaps they could comment on the compatibility of their approach with this modification of DNA-PAINT.
9. I appreciate that whilst the approach is described throughout the main text, the manuscript would benefit with the addition of a concise methods section (probably with reference to Figure 1) on how the denoising analysis steps were taken and should be included either in the methods or as a supplementary.

(Remarks on code availability)

The current linked repository only contains the readme and license. I assume it will be populated once the manuscript is published.

Unfortunately, I was unable to test the code supplied with the initial upload due to a problem obtaining the necessary license to install the 'Add-On' 'Statistics and Machine Learning Toolbox' required to use dbscan in MATLAB.

Reviewer #2

(Remarks to the Author)

This manuscript presents a well-conceived and carefully executed method for denoising DNA-PAINT super-resolution microscopy data by utilizing the predictable binding kinetics of imager and docking strands. The manuscript is clearly written, the figures are informative, and the inclusion of example code is appreciated. I believe this work represents a valuable contribution to the SMLM and quantitative imaging community. Below are a few queries and suggestions intended to improve further the clarity, applicability, and reproducibility of the study, if addressed suitably.

1) The two-step clustering method (DBSCAN followed by k-means) is effective, particularly in crowded regions. However, DBSCAN is sensitive to the choice of search radius ϵ and in this study it was set as the median localization precision. Could the authors clarify how robust the clustering outcome is to variations in ϵ ? Was any parameter sensitivity analysis performed to validate the choice of clustering parameters?

2) The use of Gaussian template matching to determine initial centroids for k-means is interesting. Have the authors compared this strategy to other centroid initialization techniques or assessed clustering quality using established metrics such as the silhouette score?

3) Is the dark-time threshold determined empirically for each sample type? Could the authors comment on the applicability of a fixed threshold in biologically heterogeneous environments where binding kinetics may vary locally?

4) Have the authors compared the performance with other goodness-of-fit tests besides the Anderson-Darling test, such as Kolmogorov-Smirnov or Cramér-von Mises, especially in the context of small sample sizes?

5) The authors propose a minimum binding event threshold (e.g., $N \geq 8$ or 12) to reject low-confidence clusters. Would a probabilistic or likelihood-based approach offer a more flexible alternative to hard thresholding, particularly for detecting sparsely expressed proteins?

(Remarks on code availability)

The method is implemented in MATLAB and demonstrates a reasonable runtime (~3 minutes in our machine for the attached dataset). Could the authors provide performance benchmarks for larger, experimentally acquired datasets? Is the current implementation scalable to 3D or high-throughput SMLM datasets, and if not, are there plans for optimization (e.g., GPU acceleration, parallelization, or batch processing)? Additionally, the provided code would benefit from more detailed in-line comments and documentation to improve clarity and facilitate adoption by other researchers. Could the authors consider enhancing the code readability and including function-level annotations and usage examples for standard datasets (for example the csv file created by ThunderSTORM) directly?

Reviewer #3

(Remarks to the Author)

(Remarks on code availability)

Version 1:

Reviewer comments:

Reviewer #1

(Remarks to the Author)

The authors have made some small changes/additions to their manuscript in an attempt to address the comments made by myself and the other reviewer/s. However, regarding points 4-7:

4. The author's initial response to this point, asking whether the background/non-specific events observed are typically higher in drosophila samples in comparison to their U2OS cells, was surprising: "We do not have experience with U2OS cells as we mostly work with Drosophila tissues". Yet, from the text, lines 351-352 read "To demonstrate this, we imaged TOM20 labelled mitochondria in U2OS cells (Fig. 6b)."

5. I thank the authors for providing the requested blink data via a Figshare repository, it is informative to see this data for the various samples imaged. The data provided did raise an additional query for the sample described as "a negative control E-cadherin sample (nc_ecad.tif)". Upon doing a simple maximum projection of the ~200 frames of the events captured one can see the E-cadherin structure, see image (showing both blink data and maximum projection for nc_ecad.tif and ecad.tif). Do I understand correctly that the negative control E-cadherin sample provided is an imager only experiment with supposedly no docking strands present? If so, how do the authors explain this appearance?

6. Additional text has been added to the discussion regarding the effects of site-loss including a statement: 'simulations suggest that the denoising algorithm is insensitive to docking strand depletion'. Arguably, the details of this/these simulation/s should now also be added to the methods/supplemental document.

7. I appreciate that the authors say they have limited knowledge and experimental experience with cultured cells but they are demonstrating an approach to remove non-specific events and have chosen to image the mitochondria of U2OS cells. The concern is that other groups who attempt to adopt this may, similar to this demonstration, apply it to targets they are unfamiliar with or have limited control over, remove specific signal and (not the case here) over interpret their data when they attempt to quantify it. I would have expected the authors to go that extra distance, when provided the opportunity, to demonstrate that the removal of real sites with their approach is limited when conducted under optimal conditions using various sample types.

(Remarks on code availability)

Reviewer #2

(Remarks to the Author)

Satisfied with the response and revision

(Remarks on code availability)

Reviewer #3

(Remarks to the Author)

(Remarks on code availability)

Version 2:

Reviewer comments:

Reviewer #1

(Remarks to the Author)

I have reviewed the additions to the manuscript and the example data. I welcome the corrections made to the example data and the inclusion of additional written methods for the work conducted.

However, I still have a fundamental issue with this manuscript in that the denoising approach used can and appears to, in the limited data provided, remove a significant amount of 'specific signal'. And so, when the authors say: "Denoising is therefore a crucial step that removes non-specific clusters for any downstream analysis and can be combined with existing quantitative methods to obtain more accurate measurements on spatial relationships and proteins quantities" I would have to argue that the denoising approach itself can also lead to inaccurate measurements on spatial relationships and protein quantities due to the removal of real sites and therefore the approach requires further validation and refinement.

(Remarks on code availability)

Utilizing the predictable binding kinetics of DNA-PAINT to denoise super-resolution images

Nature Communications NCOMMS-25-27727-T

Reviewer 1

1. In the second paragraph of the introduction the authors talk about circumventing the problem of photobleaching. Whilst this is true the authors should introduce the concept of docking strand 'site-loss' caused by photodamage. This process can lead to a decrease in localizations over time similar to photobleaching. (see: <https://doi.org/10.3390/molecules23123165>)

Response: We thank the reviewer for pointing this out. We have now modified the text and add the corresponding reference in the second paragraph to read: “DNA-PAINT circumvents the problem of photobleaching and **under optimized imaging conditions that minimize docking strand photodamage can achieve higher labelling densities and resolution down to 5 nm**”.

2. Main Figure 1. The clustering step 1 and 2 boxes would benefit from additionally having “DBSCAN” and “k-means” mentioned within them. It would also be my preference to have greater detail in the figure legend.

Response: We agree with the Reviewer. The figure has been updated, and we have expanded the figure legend.

3. Main Figure 4a. Scale bar and/or define the expected size of the nanoruler in the legend.

Response: We thank the reviewer for pointing this out. We have now added the expected size of the nanoruler to the legend.

4. Main Figure 4d. The background/non-specific events you show here appears alarmingly high. Do you typically see higher levels of non-specific binding in your *Drosophila* samples in comparison to U2OS cells? Is 2 nM a typical imager concentration you would image with?

Response: We do not have experience with U2OS cells as we mostly work with *Drosophila* tissues. In our hands it is typical to observe these levels of non-specific

binding across a variety of tissues including egg chambers, accessory glands, ejaculatory ducts, wing discs and embryos. Typical imager strand concentrations range between 1-3 nM, with 2 nM being most common.

5. *Additional media files (a hundred frames or so) that show the raw 'blinks' from binding/non-specific events in your various samples would be appreciated.*

Response: We have added 100 frames showing the raw “blinks” when imaging the following samples in a figshare repository: E-cadherin (ecad.tif), a negative control E-cadherin sample (nc_ecad.tif), mitochondria sample (mito.tif) and a-PKC sample (apkc.tif).

6. *The authors should discuss the combination of imaging sparsely distributed super-resolution targets and how the effects of DNA-PAINT site-loss could affect their denoising approach.*

Response: For a single docking strand, the denoising method will work if the docking strand is damaged after the number of binding events has passed the threshold, as this won't affect the exponentiality in the distribution of the times between binding events. For clusters with several docking strands, the effect of docking strand loss on the denoising is more complex, as it depends on several parameters including the number of docking strands in a cluster, the depletion rate of docking strands and the total number of binding events. Blumhardt et al. (2018) observed that less than 10% of the docking strands are destroyed after more than an hour of DNA-PAINT imaging in the presence of PCD and PCA. To get an estimate on the performance of our denoising approach under these imaging conditions, we ran simulations on 5000 clusters with 10 docking strands per cluster, assuming average binding time of 1 sec, average time between binding events for a single docking strand of 1000 sec, total acquisition time of 10000 sec (~1.3 h) and average lifetime of a docking strand of 80000 sec. Under these conditions each cluster lost on average ~1 docking strand, i.e 10% of the docking strands were destroyed and the statistical test rejected ~1% of the clusters. Given that the significance level of the test is set at 1%, our simulation suggests that our denoising algorithm is insensitive to effects of docking strand depletion under standard imaging conditions. In a more extreme case, we reduced the average lifetime of a docking strand in the simulation by a factor of 8, so that ~60% of the docking strands are lost during imaging. Nevertheless, the statistical test only rejected ~ 6% of the clusters.

To emphasize these points we have now added the following text and the corresponding reference in the 4th paragraph in the Discussion section: *“Also, care must be taken to minimize docking strand photodamage during imaging to ensure that the number of binding events per docking strand is maximized, as this improve the power of the statistical test and increase the levels of specific signal. Under*

typical imaging conditions with less than 10% loss in docking strands during imaging⁷, simulations suggest that the denoising algorithm is insensitive to docking strand depletion. However, excessive docking strand photodamage may cause the apparent distribution of the times between binding events to deviate from exponential, resulting in the rejection of specific clusters.”

7. The comparison between raw and denoised super-resolution images in Figure 6 highlights a particular concern that the denoising procedure could remove specific events. In Figure 6b, showing the mitochondria, it is suggestive that potentially specific events have been removed from within the mitochondrial signal leaving a number of voids. Can the authors alleviate this concern?

Response: The denoising procedure compares the cluster rejection rates between regions within the mitochondria, dominated by specific signal, and regions outside the mitochondria with non-specific signal. We note that, in this case, all signals in the areas outside the mitochondria are considered as noise, including signal from off-target binding of docking strands. This signal from the off-target docking strands is indistinguishable from specific signal and may affect the levels of specific signal that are removed from regions within the mitochondria during the denoising process. This problem can be avoided by using a negative control sample that has not been exposed to docking strands but has been prepared and imaged under standard experimental conditions. The mitochondria sample was provided by a commercial supplier, and we relied on the company’s best practices to minimize off-target binding of docking strands as we did not have access to a negative control sample.

8. Have the authors tried fluorogenic DNA-PAINT (doi.org/10.1038/s41592-022-01464-9)? Perhaps they could comment on the compatibility of their approach with this modification of DNA-PAINT.

Response: We have not tried fluorogenic DNA-PAINT because: (a) we are only aware of 2 fluorogenic imager strand sequences that have been reported to work, and this limits the scope for multiplexing. (b) the binding kinetics of fluorogenic DNA-PAINT probes complicate counting applications, such as qPAINT, because an unfolded fluorogenic probe can either refold or bind to the docking strand. However, with an appropriate statistical model to describe these more complex kinetics, the denoising method could be directly applied to fluorogenic-PAINT.

We also agree with the reviewer that it would be interesting and perhaps the focus of future work to investigate if the fluorogenic probes reduce noise levels because they may remain in their folded state when binding nonspecifically and therefore produce less fluorescence.

9. *I appreciate that whilst the approach is described throughout the main text, the manuscript would benefit with the addition of a concise methods section (probably with reference to Figure 1) on how the denoising analysis steps were taken and should be included either in the methods or as a supplementary.*

Response: We have now added a concise method section, titled “Denoising analysis”, on the denoising analysis with additional information on processing times. We have also provided a more detailed annotation of the example code.

10. *The current linked repository only contains the readme and license. I assume it will be populated once the manuscript is published.*

Response: Yes, absolutely. We apologize for not making the example code publicly available on GitHub yet. We have now uploaded the latest version on GitHub.

Reviewer 1 Code Availability

1. *Unfortunately, I was unable to test the code supplied with the initial upload due to a problem obtaining the necessary license to install the 'Add-On' 'Statistics and Machine Learning Toolbox' required to use dbscan in MATLAB.*

Response: We are sorry that the required add-on was not available and hope that the reviewer was still able to read the analysis scripts and get a sense of how the example dataset would be analyzed and what to expect for the output. We appreciate that this is a limitation of using commercially available products like MATLAB. If it were possible to use a more accessible version of the DBSCAN function within MATLAB, we would be happy to do so. We have further annotated the analysis scripts with comments to help readers understand the details even if they're unable to access the DBSCAN function in MATLAB.

Reviewer 2

1. *The two-step clustering method (DBSCAN followed by k-means) is effective, particularly in crowded regions. However, DBSCAN is sensitive to the choice of search radius ϵ and in this study it was set as the median localization precision. Could the authors clarify how robust the clustering outcome is to variations in ϵ ? Was any parameter sensitivity analysis performed to validate the choice of clustering parameters?*

Response: We chose the localization precision as the search radius for the DBSCAN step to reflect the intrinsic uncertainty of the localization process. In the two-step clustering method, we have found that the final clustering outcome is mostly dependent on the size of the gaussian template used in the second k-means clustering step. This is because regardless of how big the clusters produced by the DBSCAN step are, they will eventually be partitioned into smaller clusters during the second k-means clustering step based on the size of the Gaussian template.

2. *The use of Gaussian template matching to determine initial centroids for k-means is interesting. Have the authors compared this strategy to other centroid initialization techniques or assessed clustering quality using established metrics such as the silhouette score?*

Response: The Gaussian template offers a straight-forward way to determine both the number of clusters and their initial location. Given that the locations of the local maxima in the cross-correlation with the gaussian template indicate regions with a high density of localizations in the point cloud data, we reasoned that these locations would also provide a reasonable guess for the initial locations of the k-means centroids, and we did not investigate other initialization techniques. This method produced reasonable to strong clustering as >95% of the clusters exhibit silhouette scores >0.5 for the different sample cases presented in the manuscript, as shown in the figure below. We thank the reviewer for raising this concern regarding the clustering quality of the process which we also share. We have now added the mean silhouette scores for the clusters in Extended Figure 1 and have also modified the example code to output the mean silhouette value for the k-means clustering of each cluster produced by the first DBSCAN clustering step.

3. *Is the dark-time threshold determined empirically for each sample type? Could the authors comment on the applicability of a fixed threshold in biologically heterogeneous environments where binding kinetics may vary locally?*

Response: The dark-time threshold is determined by the binding time between the imager and docking strands. Hence, one would expect that the major influence

would be the sequences of the docking and imaging strands and the imaging buffer. In our hands, we have observed good agreement between the dark-time threshold (~1 sec) and the expected binding time based on the sequence when the docking strands are conjugated to antibodies, nanobodies and the SNAP-tag. The only exception we are aware of is the Halo-tag, which exhibits ~2x longer dark-time thresholds (~2s) than expected. This may be because the docking strand is located in a pocket in HALO, which favors rebinding of the imaging strand and reduces its apparent off-rate.

Since the imager strand off-rate, and thus the dark-time threshold, are only influenced by the immediate environment of the docking strand, its sequence and the buffer, it seems highly unlikely that it will vary across the sample, however heterogeneous.

4. *Have the authors compared the performance with other goodness-of-fit tests besides the Anderson-Darling test, such as Kolmogorov-Smirnov or Cramér-von Mises, especially in the context of small sample sizes?*

Response: We considered using the Kolmogorov-Smirnov test but, like the Cramér-von Mises test, it requires the null distribution to be specified and this requires a-priori knowledge of the average time between binding events for each cluster. Since this is not possible, and the average time between binding events must be estimated from the measured data, the result of the Kolmogorov-Smirnov test is not accurate. Instead, we considered the Lilliefors and the Anderson-Darling tests. We compared these two tests on E-cadherin negative control samples and found that the Anderson Darling test is more effective in rejecting non-specific signal, especially for small sample sizes.

5. *The authors propose a minimum binding event threshold (e.g., $N \geq 8$ or 12) to reject low-confidence clusters. Would a probabilistic or likelihood-based approach offer a more flexible alternative to hard thresholding, particularly for detecting sparsely expressed proteins?*

Response: We agree with the reviewer that the hard threshold of ≥ 12 binding events is geared towards applications that involve quantification e.g. qPAINT, because one needs this number of events to perform the downstream analysis accurately. Detecting sparsely-expressed proteins is challenging especially in situations where their distribution is not uniform across the image. In this case, one could consider segmenting the image into smaller regions and calculating a local threshold from the signal in each region. This approach can increase the numbers of non-rejected clusters with a low number of binding events as the local binding event threshold will not be affected by other regions with more abundant signal and higher numbers of binding events.

Reviewer 2 Code Availability

1. *Could the authors provide performance benchmarks for larger, experimentally acquired datasets?*

Response: The slowest part of the analysis pipeline we present is the DBSCAN algorithm. On our computers, a typical PAINT dataset of approximately two million (x, y) positions takes around 40 minutes to run in full (both clustering steps). We have now added a Method's section on the Denoising analysis and included the following text to highlight the required time for this step of the Denoising analysis:

“Typical processing times for dataset with approximately two million (x, y) coordinates was 40 minutes for completion of both clustering steps on a standard PC running 64-bit Ubuntu with 32 GB RAM and an Intel i7 CPU at 3.2 GHz.”

Additionally, we have added an Excel file to the supplementary example code (Expected Processing Time.xlsx) that gives the specifications for the computer we use for data processing along with the number of (x, y) localizations and time for the analysis to run for six experimentally acquired datasets. We hope this helps readers understand the potential time cost of using this method.

2. *Is the current implementation scalable to 3D or high-throughput SMLM datasets, and if not, are there plans for optimization (e.g., GPU acceleration, parallelization, or batch processing)?*

Response: Yes, this approach is scalable to 3D since the DBSCAN, k-means and correlation with the gaussian template can accommodate a third dimension. The main bottleneck for the analysis we present is the DBSCAN algorithm. Although using GPU or other acceleration or parallelization methods to speed up DBSCAN would be fantastic, it falls beyond our computer science capabilities, and we would welcome any input from the community in this direction. We typically ran several copies of MATLAB simultaneously per PC and ran several PCs in parallel anytime we wanted to process multiple datasets quickly.

3. *Additionally, the provided code would benefit from more detailed in-line comments and documentation to improve clarity and facilitate adoption by other researchers. Could the authors consider enhancing the code readability and including function-level annotations and usage examples for standard datasets (for example the csv file created by ThunderSTORM) directly?*

Response: We agree that the example code would benefit from more annotations and in-line comments. We have reformatted the code to make it easier to read and added more annotations and comments to help the reader understand what is happening at each step. We hope that the updated level of annotation is sufficient for a typical MATLAB user. We have also added a section to the "ExampleCode.m" file that may be enabled (uncommented) and used to select and import data from a ThunderSTORM CSV file directly. We based the import on a CSV file generated by the latest (June 2025) ThunderSTORM plugin by running the ThunderSTORM analysis on the example dataset available with the publication (<https://doi.org/10.1093/bioinformatics/btu202>).

Reviewer 3

Thank you for working with one of the other reviewers as part of this excellent initiative to facilitate training in the peer review process. We appreciate the time you spent working on these reviews and we hope the process was beneficial for you!

Utilizing the predictable binding kinetics of DNA-PAINT to denoise super-resolution images

Nature Communications NCOMMS-25-27727-T

Reviewer #1 (Remarks to the Author):

The authors have made some small changes/additions to their manuscript in an attempt to address the comments made by myself and the other reviewer/s. However, regarding points 4-7:

4. The author's initial response to this point, asking whether the background/non-specific events observed are typically higher in drosophila samples in comparison to their U2OS cells, was surprising: "We do not have experience with U2OS cells as we mostly work with Drosophila tissues". Yet, from the text, lines 351-352 read "To demonstrate this, we imaged TOM20 labelled mitochondria in U2OS cells (Fig. 6b)."

Response: The U2OS data in our manuscript comes from a commercial sample purchased from Massive Photonics. We are not working with U2OS cells and therefore we cannot comment on typical levels of background/non-specific binding events for this type of sample. We used this sample to demonstrate how the denoising method can be applied by using areas of the image outside the mitochondria to determine the non-specific cluster rejection rates instead of a negative control sample, which in this case was not available to us. For this demonstration, we relied on Massive Photonics's protocols to minimize non-specific binding events and off-target docking strand binding. We discuss the limitations of this approach and the implications it can have on the removal of specific signal given that the signal from off-target docking strands is indistinguishable from specific signal.

5. I thank the authors for providing the requested blink data via a Figshare repository, it is informative to see this data for the various samples imaged. The data provided did raise an additional query for the sample described as "a negative control E-cadherin sample (nc_ecad.tif)". Upon doing a simple maximum projection of the ~200 frames of the events captured one can see the E-cadherin structure, see image (showing both blink data and maximum projection for nc_ecad.tif and ecad.tif). Do I understand correctly that the negative control E-cadherin sample provided is an imager only experiment with supposedly no docking strands present? If so, how do the authors explain this appearance?

Response: We are grateful to the referee for pointing out this mistake. The nc_ecad.tif data was incorrectly described as the negative control. These data are actually from a *Drosophila* egg chamber containing clones of cells expressing E-cadherin-GFP next to cells that do not express E-cadherin-GFP. While the non-expressing cells serve as the negative control, E-cadherin-GFP in the expressing cells gives specific signal, which is why the cell borders appear only in certain parts of the image as the reviewer pointed out. The correct description from the nc_ecad.tif data is “E-cadherin clone sample”. We have also uploaded data for a real negative control sample and clearly labelled it “negative_control.tif”.

6. Additional text has been added to the discussion regarding the effects of site-loss including a statement: ‘simulations suggest that the denoising algorithm is insensitive to docking strand depletion’. Arguably, the details of this/these simulation/s should now also be added to the methods/supplemental document.

Response: We have now added a “Docking strand depletion simulations” section in the Methods section.

7. I appreciate that the authors say they have limited knowledge and experimental experience with cultured cells but they are demonstrating an approach to remove non-specific events and have chosen to image the mitochondria of U2OS cells. The concern is that other groups who attempt to adopt this may, similar to this demonstration, apply it to targets they are unfamiliar with or have limited control over, remove specific signal and (not the case here) over interpret their data when they attempt to quantify it. I would have expected the authors to go that extra distance, when provided the opportunity, to demonstrate that the removal of real sites with their approach is limited when conducted under optimal conditions using various sample types.

Response: The amount of specific signal erroneously removed by the denoising method for a given significance level depends on the threshold of minimum number of binding events per cluster. The optimal value of this threshold is sample dependent and to minimize the erroneous removal of specific signal in various imaging applications we chose representative samples to demonstrate different approaches on how to set this threshold e.g. with or without a negative control sample, and illustrate the strengths and limitations in each case. These approaches and the corresponding discussion are applicable to all other sample types, beyond the ones used here as examples, and will enable other researchers to choose the best approach for their work and guide them for the correct interpretation of their data.

Utilizing the predictable binding kinetics of DNA-PAINT to denoise super-resolution images

Nature Communications NCOMMS-25-27727-T

Reviewer #1 (Remarks to the Author):

I have reviewed the additions to the manuscript and the example data. I welcome the corrections made to the example data and the inclusion of additional written methods for the work conducted.

However, I still have a fundamental issue with this manuscript in that the denoising approach used can and appears to, in the limited data provided, remove a significant amount of 'specific signal'. And so, when the authors say: "Denoising is therefore a crucial step that removes non-specific clusters for any downstream analysis and can be combined with existing quantitative methods to obtain more accurate measurements on spatial relationships and proteins quantities" I would have to argue that the denoising approach itself can also lead to inaccurate measurements on spatial relationships and protein quantities due to the removal of real sites and therefore the approach requires further validation and refinement.

Response: We understand the reviewer's concern. The Denoising method is based on a statistical test which is expected to become more efficient with increasing amounts of data, in this case the number of binding events per cluster. Therefore, setting the threshold of binding events per cluster involves striking a balance between an image still corrupted by noise and, as the reviewer pointed out, an image having excessive real signal removed. The proposed Denoising method determines this optimal threshold value with the help of an image, or areas of an image, that are expected to be void of any real signal and contain only noise. Experimentally this can be achieved by using either a negative control sample or areas of the same image that are expected to lack specific signal. In the previous version of the manuscript, we only showed examples in cultured cells of the latter using mitochondria specific and non-mitochondria specific areas of an image. This approach is blind to off-target binding of docking strands and as the reviewer pointed out, can lead to the removal of non-trivial amounts of specific signal. To rectify this, we now explicitly mention this limitation in the revised manuscript. Additionally, we now describe in more detail the negative control approach, which does not suffer from the problem of off-target docking strands, and demonstrate its application by imaging microtubules in HeLa cells.